# ADAPT-∞: SCALABLE CONTINUAL MULTIMODAL INSTRUCTION TUNING VIA DYNAMIC DATA SELECTION

**Adyasha Maharana**[*]     **Jaehong Yoon**[*]     **Tianlong Chen**     **Mohit Bansal**

Department of Computer Science, UNC Chapel Hill

## ABSTRACT

Visual instruction datasets from various distributors are released at different times and often contain a significant number of semantically redundant text-image pairs, depending on their task compositions (*i.e.*, skills) or reference sources. This redundancy greatly limits the efficient deployment of **continually adaptable** multimodal large language models, hindering their ability to refine existing skills and acquire new competencies over time. To address this, we reframe the problem of lifelong Instruction Tuning (LiIT) via data selection, where the model automatically selects beneficial samples to learn from earlier and new datasets based on the current state of acquired knowledge in the model. Based on empirical analyses that show that selecting the best data subset using a static importance measure is often ineffective for multi-task datasets with evolving distributions, we propose Adapt-∞, a new multi-way and adaptive data selection approach that dynamically balances sample efficiency and effectiveness during LiIT. We first construct pseudo-skill clusters by grouping gradient-based sample vectors. Next, we select the best-performing data selector for each skill cluster from a pool of selector experts, including our newly proposed scoring function, *Image Grounding score*. This data selector samples a subset of the most important samples from each skill cluster for training. To prevent the continuous increase in the size of the dataset pool during LIT, which would result in excessive computation, we further introduce a cluster-wise permanent data pruning strategy to remove the most semantically redundant samples from each cluster, keeping computational requirements manageable. We validate the effectiveness and efficiency of Adapt-∞ over a sequence of various multimodal instruction tuning datasets with various tasks, including (Knowledge) VQA, multilingual, grounding, reasoning, language-only, and multi-image comprehension tasks. Training with samples selected by Adapt-∞ alleviates catastrophic forgetting, especially for rare tasks, and promotes forward transfer across the continuum using only a fraction of the original datasets.[1]

## 1 INTRODUCTION

Multimodal instruction tuning (Liu et al., 2023b; Zhang et al., 2023a; Liu et al., 2023a; Gan et al., 2024; Yoon et al., 2024) has been actively explored to enhance visual reasoning or the generation ability of Multimodal Large Language Models (MLLMs) (Zhu et al., 2023; Li et al., 2023a; Tang et al., 2023; Team et al., 2023; Munasinghe et al., 2023; Yu et al., 2024; Zhang et al., 2024) following user intents by training models on human or machine-generated multi-task visual instruction tuning datasets (Li et al., 2023c; Yin et al., 2023; Xu et al., 2024; Chen et al., 2024). While many distributors continue to release new high-quality instruction-tuning tasks and datasets, continually adapting large models to these massive multi-task datasets over time is prohibitively costly and inefficient. Given a pre-trained MLLM and the *continuous expansion of the dataset pool* with a stream of multi-task instruction-tuning datasets, as commonly observed in the research community today, the challenge lies in developing an ever-evolving, instruction-following MLLM in **the most data- and computation-efficient manner**. This research question poses a realistic, sustainable instruction

---

[*]Equal contribution.
[1]Code is released at https://github.com/adymaharana/adapt-inf.

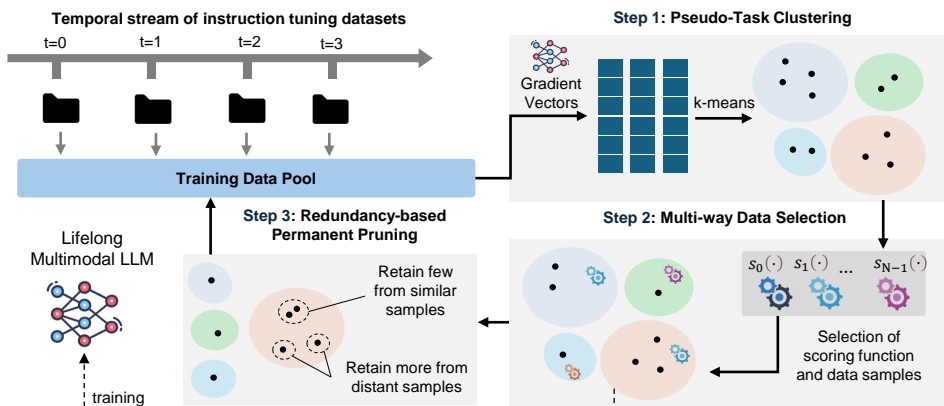

Figure 1: **Illustration of Adapt-∞.** When a new dataset is incorporated into the data pool at the beginning of each timestep, Adapt-∞ extracts sample vectors and forms pseudo-task clusters based on their similarity. Using a set of scoring functions, Adapt-∞ predicts the most suitable scoring function for each cluster and trains an MLLM on the selected samples. To prevent excessive computation as the pool size grows, we introduce dataset compression by permanently removing redundant samples.

tuning scenario for MLLMs, distinct from conventional continual learning (Zenke et al., 2017; Yoon et al., 2018; Van de Ven & Tolias, 2019), which focuses on learning a sequence of disjoint tasks. Specifically, at each time step, we assume a new multimodal multi-task instruction-tuning dataset is added to the existing training pool, which already contains previous datasets. The model aims to train on samples from this continually expanding dataset pool, a scenario we refer to as *Lifelong Instruction Tuning (LiIT)*. This scenario is critical as recent datasets tend to be exhaustively large to cover a variety of skills in uni- and multimodal tasks. It poses the unique challenges of (1) data redundancies over time due to significant overlap between datasets, (2) imbalanced task representations due to over-representation of simple tasks like captioning, (3) learning rare tasks, and (4) learning new output modalities.

Our initial experiments with *sequential* multimodal instruction tuning, *i.e.*, training on a sequence of instruction tuning datasets, show catastrophic forgetting, particularly when new output modalities such as bounding boxes and key points are introduced. While experience replay on a small subset of past datasets helps mitigate forgetting, it is insufficient for retaining rare or unique tasks that appear only once due to the skewed task distribution. This challenge is further compounded by the loosely defined nature of 'tasks' in instruction tuning (*e.g.*, multi-turn conversations across multiple tasks on the same image) and the lack of sample-wise task labels. To address this, we explore LiIT from a data selection perspective, enabling the model to learn skills in a balanced manner over time and avoid overfitting to dominant tasks. Specifically, we explore how to prune both past and incoming datasets to create a balanced training set at each time step, considering the model's current state.

To build an efficient, lifelong-evolving MLLM, we introduce *Adapt-∞: Adaptive Multi-way Pruning*, an efficient and dynamic multimodal data selection strategy. At each time step, Adapt-∞ selects the most beneficial samples for the current model from the data pool, adapting to the model's evolving knowledge and changing dataset distributions. This is crucial for LiIT, as sample importance shifts over time. Adapt-∞ operates in two major steps: (1) **Task-based clustering.** When introducing a new dataset, we integrate it into the training data pool and create pseudo-task clusters using gradient vectors to represent data samples. (2) **Cluster-wise data selection.** We select influential samples from each cluster for model training. Motivated by the observation that different score functions (Paul et al., 2021; Toneva et al.; Coleman et al.) define sample importance differently, we propose a new **multi-way data selection approach** that chooses the best scoring function from a pool of experts based on its discriminativeness (measured by entropy). It selects a skill-balanced subset of highly influential samples from the current training data pool. In addition, to better assess the influence of multimodal samples, we also propose a new scoring function, *image grounding score* (IG), which measures the change in sample perplexity when grounded by visual information. This metric prioritizes samples that enhance MLLM multimodal skills and serves as an effective data selector in Adapt-∞. As the data pool grows, computing scores and representations in Adapt-∞, like other data selection strategies, becomes costly. To ensure diversity while managing computational costs, Adapt-∞ adds (3) **Permanent data pruning** that removes semantically redundant samples from the data pool at the end of each time step, thereby continually controlling its size during LiIT.

We design an experimental setup for this previously unexplored scenario of lifelong multimodal instruction tuning using the pre-trained LLaVA 1.5 (Liu et al., 2023a) model on a stream of five visual instruction tuning datasets. As discussed earlier, experience replay alone is insufficient to mitigate forgetting, however, selecting a random subset from the data pool of past and new datasets reduces the forgetting rate from 26% to nearly 2%. Score-based data-selection strategies largely fail in this setting due to their inability to select task-balanced data subsets from multi-task datasets. Adapt-∞ not only minimizes the forgetting to a mere 0.9% using only a fraction of the training data pool, but also promotes forward transfer of skills and consistently achieves >100% relative gains.

We conduct extensive ablations of Adapt-∞ to identify best-performing settings. Our key finding is that hidden layer outputs capture semantics, while gradient vectors represent skills, making them more effective for pseudo-task clustering (see examples in Figures 2 and 4). Further, we find that gradients from the middle layer of the model lead to the best performance, suggesting that skill retention could be localized to a few layers in LLMs. We also find that zero-order gradients (Hinton, 2022; Sung et al., 2024) are promising, computationally cheaper alternatives to backpropagated gradients for pseudo-task clustering (See Table 5). Lastly, skill-wise analysis shows language-only skills are easiest to retain, while multilingual multimodal skills suffer significant forgetting in LiIT.

In summary, to the best of our knowledge, we are the first to explore the realistic setting of life-long multimodal instruction tuning where the temporal stream of datasets may contain new skills, overlapping or rare tasks, and redundant samples. This scenario necessitates dynamic data selection strategies for efficient and effective learning. Our proposed method, Adapt-∞, demonstrates superior retention as well as forward transfer of skills over time.

## 2 RELATED WORK

**Multimodal Instruction Tuning Datasets.** While multimodal data, such as image-text pairs, has increased significantly, multimodal instruction-following data remains relatively scarce due to the time-consuming nature of human data collection. To address this, recent works (Liu et al., 2023b; Zhang et al., 2023a; Chen et al., 2023; Zhu et al., 2023; He et al., 2024a) have leveraged generative models to collect such data from existing image datasets. Li et al. (2023c) introduce the M3IT dataset with 2.4 million instances and 400 task instructions, translated into 80 languages. Xu et al. (2024) develop VISION-FLAN, a large-scale dataset of 187 tasks with expert-written instructions, ensuring diversity through iterative refinement. MultiInstruct (Xu et al., 2023) features 62 tasks across 10 categories, sourced from 21 open datasets.

**Continual Instruction Tuning.** In the era of multimodal LLMs, instructional datasets have raised several timely research problems. Therefore, it is crucial to develop sustainable models that can address emerging data and real-world challenges. Inspired by continual learning (Van de Ven & Tolias, 2019; Srinivasan et al., 2022; Lee et al., 2024b), a paradigm focused on enabling models to adapt to non-i.i.d., time-variant tasks, continual instruction tuning (CIT) (Chen et al., 2024; Zhang et al., 2023b) has recently been studied for (multimodal) LLMs that allows the model to adapt to multiple instruction tuning datasets sequentially without costly retraining. KPIG (He et al., 2024b) introduces a new CIT method that helps LLMs capture task-specific information and avoid overfitting general instructions by computing key-part information gain on masked parts to replay data and refine training dynamically. EProj (He et al., 2023) and Fwd-Prompt (Zheng et al., 2024) expand the CIT to the training of large multimodal models. EProj introduces new regularization and model expansion methods based on task correlations for continual instruction tuning of LMMs. Fwd-Prompt proposes a prompt-based approach that projects the prompt gradient into the residual space to minimize task interference while utilizing pre-trained knowledge, reducing negative forward transfer.

**Data Selection.** Data selection has been explored in the form of coreset selection in many works (Welling, 2009; Chen et al., 2010; Feldman et al., 2011). Uncertainty/loss/error-based methods estimate the difficulty of a sample from model confidence (Swayamdipta et al., 2020) or its training dynamics (Toneva et al.; Paul et al., 2021; Bachem et al., 2015). Zheng et al. (2023) address catastrophic accuracy drop at high pruning rates; Maharana et al. (2024) represent datasets as undirected graphs and employ message passing to select the best subset. Gadre et al. (2024) investigate data selection for CLIP Radford et al. (2021) models. Evans et al. (2024) use learnability score (Mindermann et al., 2022) to accelerate training of CLIP models.

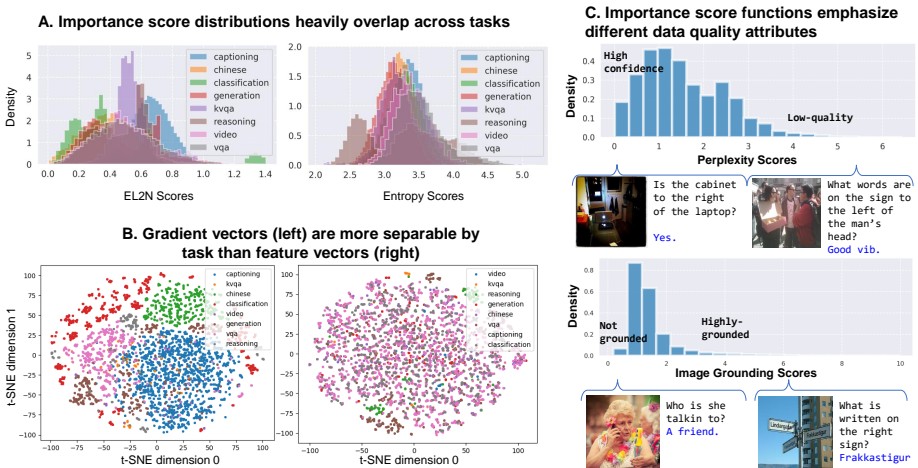

Figure 2: **A: Sample distributions for different visual language tasks** in M3IT (Li et al., 2023b) based on two importance scores, EL2N and entropy. **B: t-SNE visualization of sample vectors** based on their gradients and features. **C: Histogram** of Perplexity and Image Grounding scores. We visualize a few samples from M3IT with prompts (black) and ground-truth answers (blue)
.

# 3 LIMITATIONS OF SCORE-BASED DATA SELECTION IN LIFELONG MULTIMODAL INSTRUCTION TUNING

## 3.1 LOCALITY OF THE SAMPLE IMPORTANCE: DATA SELECTION DEPENDS ON THE DATA

Score-based selection methods are widely used to assess the importance of training samples across modalities (Zheng et al., 2023; Liu et al., 2024; Marion et al., 2023; Evans et al., 2024; Gadre et al., 2024). We analyze two importance scores in multimodal instruction tuning: the EL2N score (Paul et al., 2021) and entropy score (Coleman et al.). EL2N measures the L2-norm of the output error vector, while entropy reflects the uncertainty in the output probabilities. Using the M3IT dataset (Li et al., 2023b), which includes eight tasks: captioning, multilingual (Chinese), classification, generation, knowledge VQA (kvqa), reasoning, videoQA, and VQA, we compute score distributions. As shown in Figure 2A, relying on a single importance score metric, such as EL2N or entropy, is insufficient to differentiate meaningful samples across a diverse range of tasks. For instance, selecting higher EL2N scores tends to favor for generalization (Paul et al., 2021), *captioning* samples over *kvqa*, leading to a skewed dataset.

In addition, the effectiveness of different importance scores varies based on the task and dataset at hand. The perplexity score is effective for filtering out low-quality samples in VQA that generally occur in the tail end of its distribution (see Figure 2C). Tasks such as *kvqa* (in purple) and *captioning* (in blue) are more separable via their entropy scores than EL2N scores. Moreover, Zheng et al. (2023); Swayamdipta et al. (2020) show that the most effective training subset contains a balanced mix of easy, difficult, and ambiguous samples. A biased score estimator may assign higher or lower scores to too many samples, making it hard to select the most effective subset. Thus, we need a different, generalizable strategy for assessing sample importance across multiple datasets during multimodal instruction tuning.

## 3.2 IMPORTANCE OF VISION-LANGUAGE DATA WITH IMAGE GROUNDING SCORE

The perplexity score measures the likelihood of a given sequence of tokens as predicted by an autoregressive generative model and has been used for selecting samples with higher instruction following difficulty in language datasets (Li et al., 2024). For MLLMs, the perplexity function additionally conditions on the input image tokens within the model. A lower perplexity score implies that the model assigns a higher probability to the output tokens. We compute the perplexity of a

multimodal data instance $z_i$ for an MLLM with weights $\theta$ as:

$$\text{PPL}(z_i) = \exp\left(\frac{1}{|z_i|} \sum_{e_j \in z_i} \text{NLL}(e_j)\right), \quad \text{where } \text{NLL}(e_j) = -\log(e_j | e_{<j}, I; \theta). \quad (1)$$

where $\text{NLL}(e_j)$ indicates the negative log-likelihood of token $e_j$ in the sequence $z_i$ comprising of image $I$ and tokens $e$. As discussed in the previous section, perplexity is useful for detecting low-quality multimodal samples (see Figure 2C top). Motivated by the effectiveness and generalizability of the perplexity score (Marion et al., 2023; Li et al., 2024), we further modify this scoring function to distill the importance of the image in a multimodal data instance. We compute the image grounding score of a multimodal data instance $z_i$ with image $I$ and tokens $e$ as:

$$\text{IG}(z_i) = \frac{\text{PPL}(e)}{\text{PPL}(e, I)} \quad (2)$$

A higher IG score is assigned when the model assigns a higher probability to the text when conditioned on the image, compared to when the image is absent. Conversely, a lower IG score indicates that the image has little to no effect on the text's probability. As shown in Figure 2C bottom, an image-query pair with a higher IG score requires the model to carefully understand the visual scene (e.g., reading the text on a sign in the image). In contrast, examples where the model can predict the answer without seeing the images represent lower IG scores. Thus, the IG scoring function allows us to discard multimodal samples that do not leverage the multimodal functionality of MLLMs.

## 4 LIFELONG MULTIMODAL INSTRUCTION TUNING VIA MULTI-WAY DATA SELECTION

### 4.1 PROBLEM STATEMENT

This paper tackles the problem of LiIT over a sequence of multiple large datasets. Let $\mathcal{D}_0, ..., \mathcal{D}_{T-1}$ be a set of accessible datasets where $\mathcal{D}_t = \{x_i^t, p_i^t\}_{i=1}^{N_t}$ denotes the dataset released in the timestep $t$, composed of $N_i$ image-text pairs. Formally, we aim to train a multimodal model over multiple observed datasets for a given computational budget, such as FLOPs or training iterations. Given the model $f$ parameterized by $\theta$, the training objective at time step $T$ is formulated as follows:

$$\arg\min_{\theta} \frac{1}{T+1} \sum_{t=0}^{T} \sum_{i=0}^{\widehat{N}_t - 1} \mathcal{L}\left(f\left(\hat{x}_i^t, \hat{p}_i^t; \theta\right), \hat{y}_i^t\right) \quad \text{s.t.} \quad T \cdot (\widehat{N}_t - 1) \leq \tau, \quad (3)$$

where $\widehat{N}_t = r(N_t, T, \tau) \in \mathcal{N}$, $\widehat{N}_t \leq N_t$, and $\tau$ is the computational budget. $r(\cdot)$ denotes a decay function conditioned on $T$ and $\tau$, and $\hat{y}$ indicates the ground truth answer corresponding to the input data sample. Here, we constrain the minibatch iterations for multimodal instruction tuning per training timestep, by subsampling $\widehat{\mathcal{D}} = \{\hat{x}_i, \hat{p}_i\}_{i=1}^{\widehat{N}}$, where $\widehat{\mathcal{D}} \subseteq \mathcal{D}$, $\widehat{\mathcal{D}} \sim P(\widehat{\mathcal{D}} \mid \mathcal{D}, T, \tau)$. When a new multimodal instruction tuning dataset $D_T$ is released, $\{\widehat{\mathcal{D}}_t\}_{t=0}^{T}$ is (re-) drawn for finetuning $\theta$.

We propose the Adapt-$\infty$ data selection method for the problem of lifelong multimodal instruction tuning and describe each of its steps in detail in the following sections.

### 4.2 PSEUDO-TASK CLUSTERING VIA GRADIENTS

In the LiIT scenario, the model continuously updates its weights to incorporate new knowledge and refine its capabilities in specific tasks by training on unseen, meaningful data. The relative importance of each data sample evolves with changes in the model's state and expansion of the data pool at each time step. Therefore, adjusting the relative importance of samples within the data pool over time is crucial to faster and better optimization under restricted conditions. We accomplish this by first using gradient vectors from the model's current state to estimate skill clusters within the training data pool. As shown in Figure 2B and Figure 4, we find that gradient vectors are significantly more separable by skills than hidden state outputs of the model.

For a model with weights $\theta_l$ for layer $l$, we compute the gradients of $\theta_l$ for a sample $(z_i, y_i) = ((x_i, p_i), \hat{y}_i)$ using backpropagation. We obtain the data representation for the sample by concatenating weight gradient vectors $\nabla_{\theta} z_i = [\nabla_{\theta_0} z_i; \nabla_{\theta_1} z_i; ...; \nabla_{\theta_{L-1}} z_i]$, where $L$ denotes the number

of layers, and construct pseudo-task clusters in the data pool by performing $k$-means clustering over data representations of the seen and unseen datasets (see Figure 1). However, in practice, we find that not all gradients are necessary for distinguishing samples into multiple meaningful skills. Following Yoon et al. (2022), we cluster samples based on the gradients of essential layers, which leads to better performance compared to using gradients from all layers and is more memory-efficient as well (see discussion in Section 6).

### 4.3 Entropy-based Multi-way Data Selection

As discussed in Section 3.1, different scoring functions provide distinct advantages when selecting meaningful samples from a wide range of vision-language instruction tuning tasks, such as in-domain VQA, open-ended VQA, multilingual QA, visual grounding, commonsense reasoning, scientific/physical understanding, and OCR. To address the limitations of relying on a single data selection (pruning) method and promote collaborative decision-making when evaluating sample importance, based on the model's current state and data diversity, we propose a novel and versatile, multi-way data pruning strategy. First, we construct a function pool $\mathcal{S} = \{s_0(\cdot), ..., s_{N-1}(\cdot)\}$ where $s_n(\cdot)$ denotes $n$-th sample selection operation based on the corresponding importance score function. Here, we aim to identify the function $\hat{s}$ that maximizes the entropy of the distribution of scores over the samples of each pseudo-task cluster. For the $k$-th cluster $\mathcal{C}_k$ with $m$ samples, we first extract the corresponding scores for each $s_n$ using model weights $\theta$ at time step $t$. We approximate the distribution of scores, $P_n^\theta$, by binning the range of normalized scores and calculating densities $\hat{p}_n^b$ over the $\mathcal{B}_n^{(k)}$ bins. Omitting cluster index $k$ for brevity, the selection of $\hat{s}$ is formulated as:

$$\hat{s}^{(k)} = arg \max_{s_n} H\left(\hat{\boldsymbol{P}}_n^\theta\right), \quad \text{where} \quad \hat{\boldsymbol{P}}_n^\theta = \{\hat{p}_n^b(x)\}, \ \ \forall \boldsymbol{b} \in \mathcal{B}^{(k)} \ \text{and} \ \boldsymbol{x} \in \mathcal{C}_k. \quad (4)$$

A score function that yields higher average entropy in its distribution compared to other functions indicates a better ability to assess the uncertainty of the system (i.e., the model $\theta$ at timestep $t$). Moreover, we employ the CCS (Zheng et al., 2023) sampling strategy that aims to select a balanced representation of seen and unseen samples as well as frequent and rare tasks by sampling uniformly over the range of a score function. Hence, it is even more imperative to use a score function that is discriminative over its entire range. Since the resulting pseudo-task clusters may vary in size, we define a training budget $T$ and divide it uniformly over the $k$ clusters. The leftover data budget from clusters with sizes $|c_i|$ smaller than $T/k$ are equally distributed over other clusters. This results in the selection of a skill-balanced subset from the pool of multi-task datasets in lifelong learning. Importantly, our proposed multi-way approach is highly flexible since the scoring function pool can be seamlessly extended with new scoring functions based on users' needs.

### 4.4 Combined Permanent-Temporal Data Pruning by Removing Redundancy

The steps of Adapt-$\infty$ outlined in the previous sections enable the model to effectively select the most beneficial samples for each pseudo-task (i.e., skill) and train on them adaptively, based on the current model and evolving dataset distributions. However, as the data pool grows with the release of new instruction-tuning datasets, the computational burden of updating gradient-based sample vectors and ranking their importance increases. To keep the computational requirements manageable over time, we further implement a permanent data pruning strategy to iteratively reduce the size of the entire dataset pool. At the end of training at each timestep, we measure pairwise cosine similarities between samples within each cluster and prune those with maximum similarities, as they are highly redundant and contain overlapping knowledge. The similarity is computed in the semantic space, which is well represented using hidden layer outputs of the model as shown in Figure 4 (Abbas et al., 2023; Sorscher et al., 2022). Larger clusters are prioritized for pruning to fit a predefined data pool budget $D$ until all clusters retain a uniform number of samples. We refer to the version of Adapt-$\infty$ with this combined permanent-temporal pruning as LITE-Adapt-$\infty$.

## 5 Experiments

### 5.1 Experimental Setup

**Model.** We conduct our experiments using the LLaVA 1.5 multimodal large language model (Liu et al., 2023a). It is trained on top of the Vicuna LLM (Chiang et al., 2023) using a corpus of approx-

Table 1: **Comparison of multimodal instruction tuning datasets.** Distribution of various skills in the datasets. RE: Referring Expression, OD: Object Detection, KD: Keypoint Detection.

| Training Datasets | Dataset Size | Skills | | | | | | | |
|---|---|---|---|---|---|---|---|---|---|
| | | VQA | Knowledge VQA | Captioning | Multi-lingual | Non-text Output | Video QA | Complex Reasoning | Language Only |
| LLaVA-1.5 (Liu et al., 2023b) | 665K | ✓ | ✓ | ✓ | ✗ | RE | ✗ | ✓ | ✓ |
| M3IT (Li et al., 2023c) | 2.1M | ✓ | ✓ | ✓ | ✓ | ✗ | ✗ | ✗ | ✗ |
| MiniGPT4 (Zhu et al., 2023) | 3K | ✗ | ✗ | ✓ | ✗ | ✗ | ✗ | ✗ | ✗ |
| MANTIS (Jiang et al., 2024) | 666K | ✓ | ✗ | ✗ | ✗ | ✗ | ✗ | ✓ | ✗ |
| LaMM (Yin et al., 2023) | 250K | ✓ | ✗ | ✓ | ✗ | OD & KD | ✗ | ✗ | ✗ |
| VisionFLAN (Xu et al., 2024) | 191K | ✓ | ✓ | ✓ | ✗ | ✗ | ✗ | ✓ | ✗ |

imately 600K image-text caption pairs for pretraining of the vision projectors from the pre-trained CLIP visual encoder ViT-L/14 (Radford et al., 2021). Further, it is trained on the LLaVA instruction tuning dataset consisting of 665K text-only and vision-language instances. We adopt LoRA finetuning (Hu et al., 2021) of the LLaVA-1.5-7B model with the recommended hyperparameters.[2]

**Datasets.** For training, in addition to the LLaVA-665K instruction tuning dataset at $t = 0$, we consider the following order of datasets: M3IT (Li et al., 2023c), MiniGPT4 (Zhu et al., 2023), MANTIS (Jiang et al., 2024), LAMM (Yin et al., 2023) and VisionFLAN (Xu et al., 2024). Each dataset's temporal order, size, and skill composition are summarized in Table 1. We select standard evaluation datasets to measure performance on the skills enumerated in Table 1. These datasets and their corresponding task-specific evaluation metrics are listed in Table 4.

**Metrics.** We report various evaluation metrics from existing literature designed for understanding the continual learning phenomena in machine learning. The **Average Accuracy** ($acc$) at final timestep $t = T$ is the average of the model's performance across all skills (and across datasets within each skill). The **Relative Gain** ($r$) metric (Scialom et al., 2022) is the average of skill performances as a % of the respective upper bounds i.e., $r^T = \frac{1}{S} \sum_{s=1}^{S} \frac{P_s^t}{upper\,bound^s} \times 100\%$. We consider the best performances in each skill group in the sequential learning setting to be the upper bound in performances and report $r$ for the final time step $T$. We also report the **Forgetting Rate**($f$) which is the % drop in performance averaged across all skills and timesteps i.e., $f = \frac{1}{S \times T} \sum_{s=1}^{S} \sum_{t=1}^{T} \frac{min(P_s^t - P_s^{t-1}, 0)}{P_s^{t-1}} \times 100\%$.

**Adapt-$\infty$ Setup.** The optimal value of $k$ in the pseudo-task clustering step is computed from a grid search over values of $k$ between 5 and 50, and selected based on the WSS value of clusters.[3] In the score-based sample selection step, we use a bin size of 50 and discard the top and bottom 5% of samples for computing entropy as well as for CCS sampling, to remove outliers, low-quality samples, and uninformative data (Zheng et al., 2023). We use perplexity, image grounding (Section 3.2), EL2N (Paul et al., 2021) and entropy (Coleman et al.) score functions for $\mathcal{S}$ throughout the paper.

**Baselines.** We present baseline numbers on (1) Sequential and Multi-task training, (2) Random selection for experience replay (10% of past datasets), (3) Score-based selection methods, including Random, EL2N (Paul et al., 2021), Entropy (Coleman et al.), Perplexity (Marion et al., 2023), and (3) recent competitive data pruning baselines: SemDeDup (Abbas et al., 2023), Density-based Pruning (Abbas et al., 2024), and COINCIDE (Lee et al., 2024a). See Appendix for details.

## 5.2 Main Results

We present the main experimental results in Table 2 and show a breakdown of skill-wise accuracies at each time step for various methods in Figure 3. Our main findings are as follows:

**Sequential training leads to catastrophic forgetting of skills.** Multiple skills learned at $t$=0 (LLaVA) are forgotten at $t$=1,2 upon training the model on datasets containing a different set of skills. The M3IT dataset ($t$=1) does not contain grounding tasks and results in large drops in performance for the same. Similarly, the MiniGPT4 dataset ($t$=2) predominantly contains high-quality captions and causes forgetting of all other skills. The MANTIS dataset ($t$=3) improves performance on the MMMU dataset because it contains training instances with documents, charts, and visualization images. At $t$=4, the LAMM dataset contains object detection and keypoint detection tasks that

---

[2] https://github.com/haotian-liu/LLaVA/tree/main

[3] Within the sum of squares (WSS) is sum of squared distance between samples and their cluster centroids.

Table 2: **Overall results for lifelong multimodal instruction tuning.** Comparison of performance of LLaVA models trained on datasets selected using Adapt-∞ vs. other data selection methods.

| Pruning Strategy | Data size at $t$ | Relative Gain % (↑) | Forgetting Rate % (↓) | Avg. Acc. (↑) |
|---|---|---|---|---|
| Sequential | Full | 68.0 | 26.0 | 32.0 |
| Multi-task | Full | 92.5 | - | 46.1 |
| Random Experience Replay | Full | 89.5 | 6.6 | 43.4 |
| Random | 25k | 95.3 | 2.1 | 47.2 |
| EL2N (Paul et al., 2021) | 25k | 82.4 | 12.2 | 43.9 |
| Entropy | 25k | 79.6 | 15.1 | 41.6 |
| Perplexity (Marion et al., 2023) | 25k | 91.4 | 9.6 | 45.2 |
| Image-grounding (ours) | 25k | 92.3 | 5.6 | 45.6 |
| SemDeDup (Abbas et al., 2023) | 25k | 76.4 | 6.4 | 38.5 |
| Density-based (Abbas et al., 2024) | 25k | 78.1 | 5.1 | 39.6 |
| COINCIDE (Lee et al., 2024a) | 25k | 89.5 | 3.9 | 44.7 |
| Adapt-∞ | 25k | 102.3 | 0.9 | 50.5 |
| Adapt-∞ | 50k | 107.2 | 0.2 | 51.7 |
| Adapt-∞ | 100k | **109.7** | **0.4** | **52.5** |
| LITE-Adapt-∞ | 25k | 99.7 | 1.3 | 49.6 |

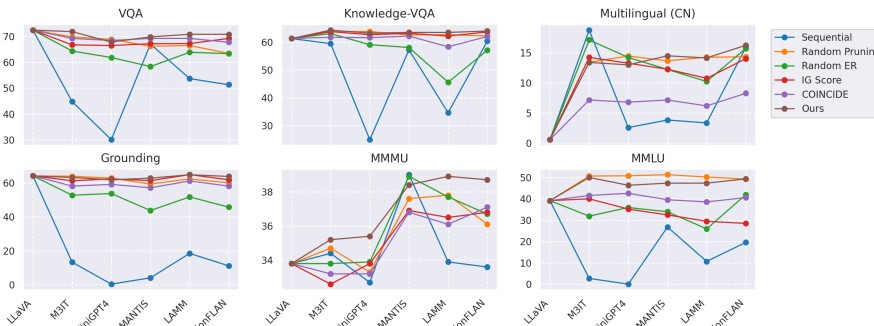

Figure 3: **Average accuracies per skill over time.** Comparison of average accuracies over time for each skill in our evaluation suite using various data selection methods. Higher is better.

promote recovery of the referring comprehension skill learned at $t = 0$, presumably due to a similarity in their non-textual output modalities. Finally, VisionFLAN ($t$=5) recovers performance on all skills except grounding and MMMU. Surprisingly, VisionFLAN also induces recovery of multilingual and unimodal skills (MMLU) despite not containing those tasks in its dataset composition. This method incurs nearly 26% forgetting across all timesteps and retains only 68% of its all-time high performances at the final timestep (see Table 2). Models trained using Adapt-∞ also outperform other models in multi-turn visual conversations (see Figs. 6, 7 and 8).

**Random experience replay and pruning alleviate forgetting.** Random experience replay using 10% data from past datasets significantly improves skills retention over sequential training, bringing down the forgetting rate from 26% to 6.6%. Random pruning of the combined set of past and incoming datasets results in better retention of skills with only a 2.1% forgetting rate using a fraction of the datasets for training as seen in Figure 3. This result demonstrates the need for applying data selection methods to the combined datasets rather than in isolation and serves as a strong baseline for lifelong multimodal instruction tuning. Notably, random pruning also improves the language-only skill of the underlying LLM in LLaVA. However, random pruning achieves a relative gain of 95%, falling short of reaching the best performance seen during sequential training.

**Adapt-∞ minimizes forgetting and promotes forward transfer.** Adapt-∞ outperforms all existing data selection methods for retaining and learning skills via instruction tuning over time. Score-based data selection methods generally fall short of random pruning because selecting samples from multi-task datasets based on scores leads to a skewed representation of tasks in the subset (see Figure 2). Our proposed scoring function, Image grounding, prioritizes the data samples where the outputs are strongly grounded in images (see Fig. 2C) and achieves relatively higher performance for multimodal tasks. SemDeDup (Abbas et al., 2023), DBP (Abbas et al., 2024), and COINCIDE (Lee et al., 2024a) rely on hidden layer outputs from the model to represent and prune data samples which can lead to imbalanced task distributions in the selected subset.

The use of gradient vectors in Adapt-∞ to pool samples into pseudo-task clusters before pruning ensures that all skills are well-represented at each time step, resulting in the lowest forgetting rate

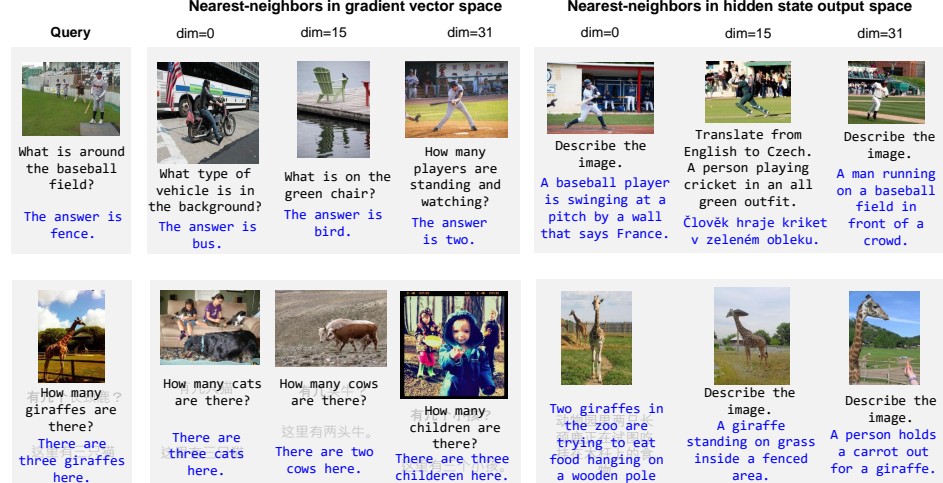

Figure 4: **Nearest neighbor samples** for query samples from VQA (top) and Chinese VQA (bottom) tasks in the gradient (left) and feature spaces (right).

Table 3: **Ablation results for Adapt-∞.** Comparison of performance of the Adapt-∞ method to its ablated versions for lifelong multimodal instruction tuning.

| Ablation Type | Values | Relative Gain % (↑) | Forgetting Rate % (↓) | Avg. Acc. (↑) |
|---|---|---|---|---|
| **Within-cluster Pruning** | *Multi-way* | **102.3** | **0.9** | **50.5** |
| | Image Grounding Score | 96.3 | 2.7 | 48.8 |
| | EL2N | 97.4 | 1.5 | 49.1 |
| **Data Representations** | *Gradients* (middle layer) | **102.3** | **0.9** | **50.5** |
| | Gradients (first layer) | 97.2 | 1.8 | 47.9 |
| | Gradients (last layer) | 101.5 | 1.3 | 50.1 |
| | Gradients (all layers) | 98.9 | 2.5 | 49.1 |
| | Hidden layer outputs (all layers) | 96.5 | 4.3 | 47.4 |
| **Cluster Budget** | *Uniform* | **102.3** | **0.9** | **50.5** |
| | Density-based | 101.7 | 0.5 | 49.5 |

i.e., 0.9%. Further, the multi-way score-based selection of *important* samples within those clusters promotes forward transfer of skills and results in >100% relative gain, unlike any other method in Table 2. The relative gain increases by nearly 5% on doubling the data budget from 25K to 50K samples and shows signs of plateauing with further increase in data.

**Semantic deduplication for managing data complexity over time is effective.** LITE-Adapt-∞ employs semantic deduplication of the training pool at the end of each timestep to reduce the computation complexity of extracting data representations at the next step. For the deduplication size of 100K samples (4x of training budget) across all timesteps, LITE-Adapt-∞ suffers minor drops in performance and forgetting as compared to Adapt-∞.

# 6 ANALYSIS

**Relative gains per skill.** We present the skill-wise breakdown of relative gains of each method discussed in Table 2, in Figure 5A. The largest increases are observed for the language-only skill across all methods, except the image-grounding score which deprioritizes unimodal samples. This result is especially striking because none of the datasets in our experimental setting contain samples similar to those in the MMLU evaluation dataset. It suggests that the underlying LLM in LLaVA can recover this skill from similar multimodal samples. The next highest gains are seen for the knowledge VQA dataset, using our image-grounding score and the Adapt-∞ method. Multilingual skills appear to be the hardest skills to learn and retain over time, potentially because they utilize a different part of the model's vocabulary than other tasks and are included in the M3IT dataset only.

**Multi-way pruning vs. single pruner.** Adapt-∞ uses one among various importance score functions (or pruner experts) for each cluster to select a representative subset of importance samples from the cluster. This works better than using any one single metric across all clusters as shown in

Figure 5: **A: Relative gain % for different skills** using various data selection methods in the lifelong multimodal instruction tuning setting. **B: t-SNE visualization of gradient vectors** of the M3IT dataset from layers of varying depth in the LLaVA model.

Table 3. Importantly, this serves as a flexible framework for swapping or adding advanced 'expert' pruners, e.g., learnability score (Mindermann et al., 2022; Evans et al., 2024).

**Source of data representations.** Various data selection works propose different ways of representing data samples in a high-dimensional space. Methods designed for pruning pretraining datasets or single-task finetuning can effectively use semantic representations such as sentence embeddings (Abbas et al., 2023) and hidden state outputs (Sorscher et al., 2022; Maharana et al., 2024). However, we observe subpar performance with the use of semantic representations for pseudo-task clustering, as reported in Table 3. Gradient vectors are better representations of the skill component of a data sample as we show in Figure 4. Further, we experimented with gradients from all layers (similar to Xia et al. (2024); Liu et al. (2024)) as well as individual layers of the model. We observed the best performance with gradients from the middle layer only. Gradients from the last layer also work relatively well with Adapt-∞ but those from the first layer work poorly. This discrepancy correlates with the compactness of task clusters in the t-SNE plots of gradients from the corresponding layers, as demonstrated in Figure 5B.

**Sampling budget across clusters.** As outlined in Section 4, we sample a subset with an equal number of samples from each pseudo-task cluster in Adapt-∞. The budgets leftover from smaller clusters are distributed equally across the remaining clusters. We experimented with more intuitive budgets for each cluster i.e., based on the density of the cluster members (Abbas et al., 2024). However, it did not result in significant changes in the performance.

**Efficiency.** Most data selection methods require extra steps to select an influential data subset and incur computational costs. Adapt-∞ expends additional memory and inference-time compute to effectively select data. We experiment with three methods to reduce this cost: (1) Zero-order gradients (Hinton, 2022), (2) Varying size of data pool in LITE-Adapt-∞, and (3) gradients from a smaller model i.e., TinyLLaVA (Zhou et al., 2024). Using zero-order gradients instead of full gradients leads to a 2% drop in average accuracy and a 2.4% drop in relative gains. TinyLLaVA gradients demonstrate a similar drop, in addition to a higher forgetting rate. Conversely, the performance of LITE-Adapt-∞ improves with increasing size of the data pool post-deduplication (see Table 5).

See Appendix C for additional results using LLMs, analysis on efficiency and recovery of skills.

## 7 CONCLUSION

Valuable visual instruction tuning datasets from various sources are released over time and often contain overlapping text-image pairs. To efficiently train lifelong adaptive MLLMs on these growing datasets, a scenario we call Lifelong Instruction Tuning (LiIT), we reformulate data selection so the model automatically selects meaningful samples from both old and new datasets, maintaining balance when incorporating new data. We observe that assessing sample informativeness with a static importance measure is challenging in LiIT, as it depends on the model's evolving capabilities and the shifting dataset distribution. To address this, we propose a scalable lifelong multimodal instruction tuning approach that dynamically balances sample efficiency and effectiveness through temporal multi-way data pruning. We show that training with samples selected by this method reduces catastrophic forgetting and enhances forward gain, using only a fraction of the original dataset, particularly for rare tasks with limited resources.

## ACKNOWLEDGEMENTS

We thank the reviewers for their valuable feedback and input on the paper. This work was supported by the National Institutes of Health (NIH) under other transactions 1OT2OD038045-01, NSF-CAREER Award 1846185, DARPA ECOLE Program No. HR00112390060, NSF-AI Engage Institute DRL-2112635, DARPA Machine Commonsense (MCS) Grant N66001-19-2-4031, ARO Award W911NF2110220, ONR Grant N00014-23-1-2356. The views contained in this article are those of the authors and not of the funding agency.

## ETHICS STATEMENT

The intended use of Adapt-$\infty$ is to enhance the vision-language instruction-following capabilities of MLLMs by training them on an integrated, ever-growing instructional dataset over time. This system does not pose any specific potential for misuse beyond the general risks associated with AI technology. However, instruction-tuned frameworks, including Adapt-$\infty$, are required to carefully consider the selection of training datasets and the intended purpose of usage to ensure the development of a reliable and trustworthy video-language inference system that supports stable, reliable, and safe AI.

## REPRODUCIBILITY STATEMENT

This paper fully discloses all the information needed to reproduce the main experimental results of the paper to the extent that it affects the main claims and/or conclusions. To maximize reproducibility, we have included our code in the supplementary material. Also, we report all of our hyperparameter settings and model details in the Appendix.

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

Table 4: **Evaluation datasets** used for measuring the performance of the trained model at each time step in our continual learning experiments.

| Skill | Evaluation Datasets | Evaluation Metric |
|---|---|---|
| VQA | GQA (Hudson & Manning, 2019), LLaVA-Bench (Liu et al., 2023a) | Accuracy |
| Knowledge VQA | ScienceQA (Lu et al., 2022), OK-VQA (Marino et al., 2019), A-OK-VQA (Schwenk et al., 2022) | Accuracy |
| Multilingual (Chinese) | COCO-CN, Flickr-8K-CN (Li et al., 2023b) | BLEU Score |
| Grounding | RefCOCO, RefCOCO+, RefCOCOg (Kazemzadeh et al., 2014; Yu et al., 2016) | IOU |
| Reasoning | MMMU (Yue et al., 2024) | Accuracy |
| Language | MMLU (Hendrycks et al., 2021a;b) | Accuracy |

# APPENDIX

## A    BASELINES

**Sequential Training.**    In this method, the model is naively trained on the stream of instruction tuning datasets without any experience replay. Generally, this method sets a lower bound on the performance of the model on each evaluation task.

**Multi-task Training.**    This method comprises pooling all of the datasets across time steps and training the model on this pooled dataset in one go. Generally, this method sets an upper bound on the performance of the model on various skill sets. However, if there are low-resource tasks in the dataset, it can lead to low performance on those tasks.

**Coverage-based Coreset Selection (CCS).**    Zheng et al. (2023) introduces the method CCS where they divide a range of difficulty scores into equal-sized bins and randomly data samples from each bin with a uniform budget. This approach is motivated by maximizing coverage over the semantic space while ensuring an equal distribution of *easy* and *difficult* samples in the selected subset. Easy samples promote learning whereas difficult samples are information-dense and promote generalization. We use this method in Adapt-$\infty$ for score-based selection.

**Score-based Selection.**    Multiple importance score functions have been proposed over the years for various data types. We select the following for baseline experiments: (1) *Perplexity*: This metric is widely used for filtering language corpora (Marion et al., 2023), (2) *EL2N* (Paul et al., 2021): This metric is the L2-norm of the output error vector and is effective at low pruning ratios (or high retention rates), (3) Entropy (Coleman et al.): This score function is the entropy value of the output probability vector.

**Embedding-based Selection.**    SemDeDup (Abbas et al., 2023) extracts semantic embeddings for pertaining datasets using a universal embedding transformer such as CLIP (Radford et al., 2021) for image-text pairs or Sentence Transformer[4] for natural language corpora and performs deduplication within $k$-means clusters. DBP (Abbas et al., 2024) assigns pruning budget to the clusters in SemDeDup using a cluster complexity score. COINCIDE (Lee et al., 2024a) clusters feature vectors into a large number of cluster e.g., $k$=10,000 to identify skill-concept clusters and samples non-uniformly from the clusters using a difficulty score metric.

---

[4]https://www.tensorflow.org/hub/tutorials/semantic_similarity_with_tf_hub_universal_encoder

Table 5: **Efficiency results**. Comparison of performance of efficient versions of the Adapt-∞ method for lifelong multimodal instruction tuning. $T$ = Training data size; $|D|$ = Size of data pool after permanent pruning in LITE-Adapt-∞.

| Method | Relative Gain % (↑) | Forgetting Rate % (↓) | Avg. Acc. (↑) |
|---|---|---|---|
| Adapt-∞ (T=25k) | 102.3 | 0.9 | 50.5 |
| + TinyLLaVA gradients (Zhou et al., 2024) | 99.6 | 2.9 | 48.1 |
| + Zero-order gradients (Hinton, 2022) | 99.9 | 1.5 | 48.5 |
| LITE-Adapt-∞ ($|D|$=100k) | 99.7 | 1.3 | 49.6 |
| LITE-Adapt-∞ ($|D|$=200k) | 101.8 | 1.1 | 50.1 |
| LITE-Adapt-∞ ($|D|$=500k) | 102.5 | 0.7 | 50.8 |

Table 6: **Efficiency analysis**. Comparison of *total time* taken for various data selection methods across all time steps in our experiments with 25k training samples at each $t$. Numbers are in hours.

| Method | Random Selection | Scoring-based | LESS(Xia et al., 2024) | COINCIDE (Lee et al., 2024a) | Adapt-∞ | LITE-Adapt-∞ |
|---|---|---|---|---|---|---|
| Data scoring | - | 48 | 92 | 48 | 92 | 34 |
| $k$-means clustering | - | - | - | 3.5 | 3.5 | 1.4 |
| Training | 21 | 21 | 21 | 21 | 21 | 21 |
| **Total** | 21 | 69 | 113 | 72.5 | 116.5 | 56.4 |

# B    EXPERIMENTAL SETUP

**Additional details on Adapt-∞ setup.** We use random projections (Park et al., 2023; Xia et al., 2024) to reduce the dimensionality of gradient vectors extracted for the pseudo-task clustering step. We use a constant projection dimension of 8192 throughout our experiments.

# C    ADDITIONAL RESULTS

**Efficiency analysis.** We present a comparison of the time taken by various data selection methods for our experimental setting (for training with 25k samples at each time step) in Table 6. Results are presented for 8 A100 GPUs. The total time taken to train the model without any methodical data selection (i.e., random pruning) is approximately 21 hours. Scoring-based selection methods generally require a forward pass of the model to extract the score (e.g., EL2N, entropy), which takes nearly 48 hours on 8A100 GPUs for all datasets in our experiments. The COINCIDE (Lee et al., 2024a) takes a similar amount of time since it uses a forward pass to extract hidden layer outputs from the models as data representations. LESS (Xia et al., 2024) and our proposed method Adapt-∞ require longer time i.e., 92 hours, to perform a backward pass over the model and extract gradients. However, with LITE-Adapt-∞, we are able to significantly reduce this time due to systematic compression of the dataset at each timestep.

**Ablations with scoring functions.** We present ablation results for the different types of scoring functions used in Adapt-∞. The results presented in Table 2 for Adapt-∞ include four scoring functions as outlined in Sec. 5.1 i.e., perplexity, image grounding (Section 3.2), EL2N (Paul et al., 2021) and entropy (Coleman et al.) score functions. We refer to this set of functions as $\{\mathcal{S}\}$. We performed additional experiments where we used (a) only the image grounding score as the scoring function, and (b) all scoring functions except the Image-grounding score. Results are reported in Table 7. We find that the image grounding score serves as a good scoring function when used as a standalone scorer in Adapt-∞. In addition, Adapt-∞ performs well even when the image grounding score is removed from the set of best performing score functions. This suggests that performance with our method can improve with better scoring functions, making Adapt-∞ a versatile method.

**Visual chat skill.** Multimodal LLMs are equipped with chat-style multi-turn conversational skills due to the chat capabilities of the underlying LLM as well as instruction tuning data tailored to the skill. The LLaVA instruction tuning dataset contains such data; hence, the LLaVA model can hold multi-turn chats with a user. However, the multi-turn visual chat task is not well-defined and lacks quantitative evaluation metrics or benchmarks. Nevertheless, we are interested in seeing if

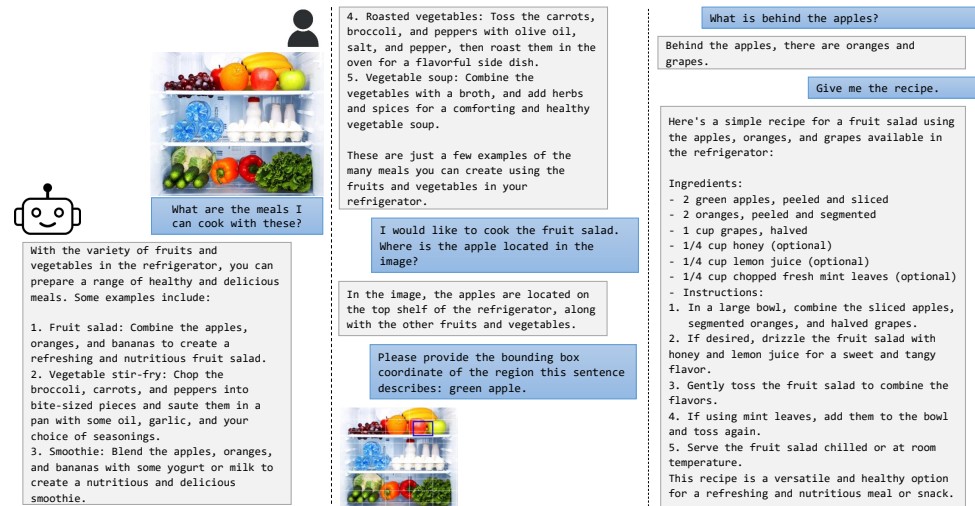

Figure 6: **Visual chat with LLaVA at $t$=0 (Liu et al., 2023a).** We use a representative chat example to evaluate the visual chat capability of MLLMs trained using various methods in our experiments.

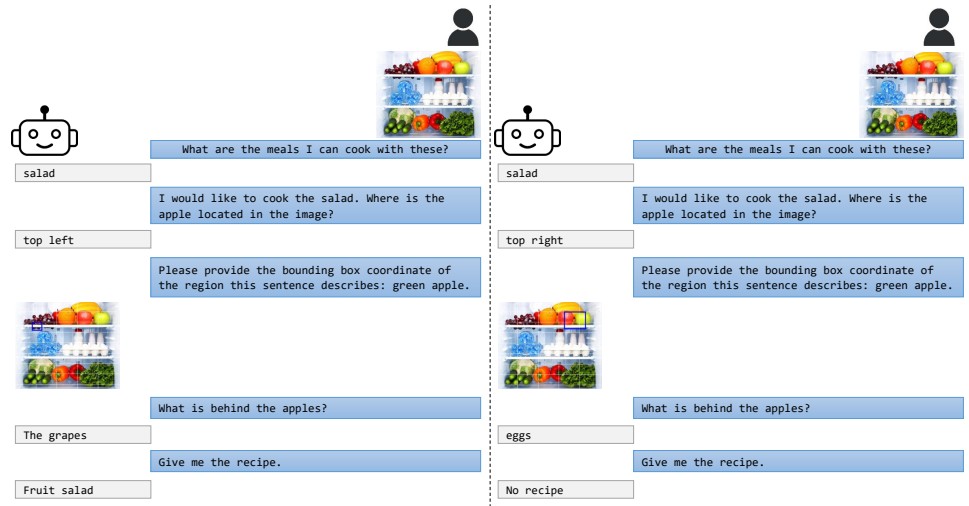

Figure 7: **Visual chat with LLaVA at $t$=5** using sequential training (left) and random pruning (right) baselines in our experiments. The conversational ability of the LLaVA model is greatly diminished over time using these baseline data selection methods.

Table 7: **Ablations results for scoring functions in Adapt-∞ for multimodal models.** Comparison of performance of LLaVA model trained on datasets selected using Adapt-∞ under different combinations of scoring functions.

| Scoring Functions | Relative Gain % (↑) | Forgetting Rate % (↓) | Avg. Acc. (↑) |
|---|---|---|---|
| $\{\mathcal{S}\}$ | 109.7 | 0.4 | 52.5 |
| $\{\mathcal{S}\}$ - Image Grounding Score | 106.9 | 0.3 | 51.2 |
| Image Grounding Score *only* | 92.3 | 5.6 | 45.6 |

the multi-turn chat skill is retained in the LLaVA model during lifelong learning, especially since none of the fine-tuning datasets, except for LLaVA-665K, contain conversational data. We perform qualitative evaluation of the visual chat skill using a single representative example demonstrated in

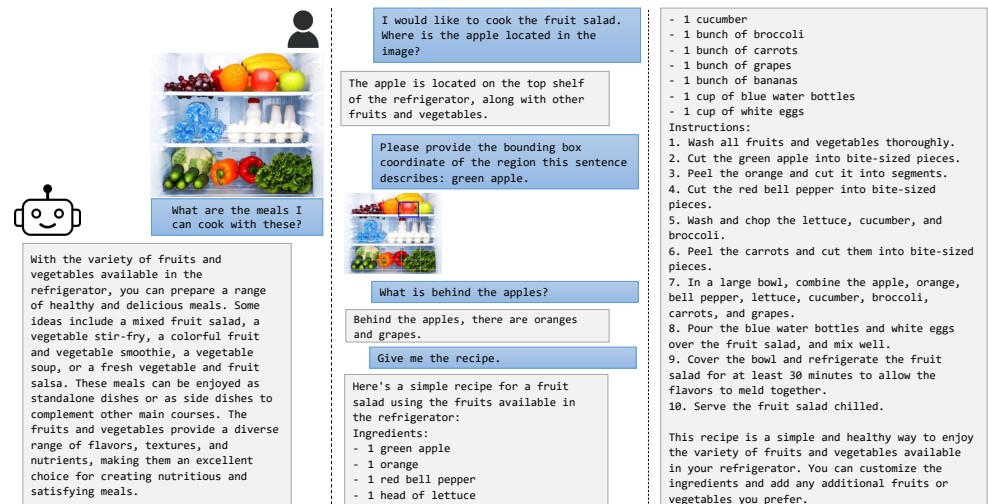

Figure 8: **Visual chat with LLaVA at $t$=5** using our proposed dynamic data selection method Adapt-$\infty$. Unlike other methods (see Fig. 7), the Adapt-$\infty$ method identifies conversation as a distinct task in the pseudo-task clustering step and retains sufficient samples from this task at each time step to prevent forgetting the skill.

Fig. 6. In this example, the model is queried for VQA, knowledge VQA, and referring expression comprehension tasks based on a single image. As seen in Fig. 6, the LLaVA model (at $t$=0 in our experiments) can perform all these tasks effectively in a multi-turn chat scenario, except for fine-grained visual reasoning. The models trained using sequential learning and random pruning in our experiments lose this skill, as shown in Fig. 7. The models' answers become less verbose and they provide inaccurate answers for most tasks. We present results from our method in Fig. 8. Adapt-$\infty$ identifies the multi-turn chat as a distinct task during the pseudo-task clustering step and selects samples from this cluster for training at each time step. Thus, it can retain this skill effectively even after many steps of fine-tuning on other datasets.

**What is important for multimodal skill recovery?** As noted in Section 5.2, a model can recover previously learned skills *without* being re-trained on data from that skill. Sequential training of the LLaVA model in our experimental setting shows that the model at $t$=4 has poor multilingual skills (introduced at $t$=1), and recovers those skills at $t$=5 without being trained on multilingual data (See the result of *Sequential* on *Multilingual (CN)* task in Figure 3). We use this phenomenon to understand what aspects of the training data induce skill recovery in a multi-

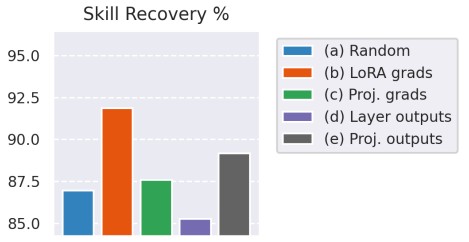

Figure 9: **Recovery of multilingual skill** with various training data selection methods.

modal model. To this end, we train the model from $t$=4 timestep on different subsets of the Vision-FLAN dataset at $t$=5 and evaluate its performance on the multilingual skill. Specifically, in $t$=5, we train the model on an equal number of (a) randomly selected samples, most similar samples in (b) LoRA gradient space, (c) vision projection layer gradient space, (d) hidden layer output space and (e) projection layer output space. We compute the average cosine similarity of each sample in the VisionFLAN dataset with the multilingual subset of the M3IT dataset ($t$=1). Gradients and outputs are extracted from the vision projector layers and the decoder layers to represent the visual and multimodal content of the samples, respectively. We report the relative gain of the trained model as a measure of skill recovery. As presented in Figure 9, the best multilingual performance is achieved by (b) *training on VisionFLAN samples that have similar gradients in the LoRA modules*, followed by (e) samples with similar outputs from the visual projection layer. Since we have already established that gradients represent the skill whereas hidden layer outputs represent the semantics (see Section 4.2), these results suggest that multimodal models can retain as well as recover previously

Table 8: **Overall results for lifelong instruction tuning in language models**. Comparison of performance of LLaMa3-8B models trained on datasets selected using Adapt-∞ vs. other data selection methods.

| Method | Relative Gain % (↑) | Forgetting Rate % (↓) | Avg. Acc. (↑) |
|---|---|---|---|
| Sequential | - | 18.1 | 36.8 |
| Random | 91.6 | 5.6 | 41.8 |
| LITE-Adapt-∞ | 97.8 | 2.3 | 43.1 |

learned skills through reminding indirect representations of the skill in training datasets. We also observe this phenomenon at $t$=4 in the sequential setting where training the LLaVA model on object and keypoint detection tasks leads to recovery of the referring comprehension skill. These results provide further evidence for the effectiveness of gradient-based clustering of datasets in Adapt-∞.

**Data selection for language-only models.** We performed experiments for lifelong instruction tuning (LiIT) of language models to explore the efficacy of our proposed approach for language-only data. For this purpose, we use the LLaMA3-8B (Dubey et al., 2024) model and five multi-task language datasets in the following training order: Natural Instructions (Mishra et al., 2022), Alpaca-CoT (Qingyi Si, 2023), xP3 (Muennighoff et al., 2022), UltraChat (Ding et al., 2023) and FireFly.[5] We use the set of evaluation benchmarks used in Wang et al. (2023) to evaluate our models (MMLU, GSM, BBH, TydiQA, CodexEval, AlpacaEval) and compute the following metrics based on performance across these benchmarks: Average accuracy, Relative gain, and Forgetting rate (see Sec. 5.1 in main text). The model is trained on 25k samples at each time step, similar to the MLLM experiments in our paper. We report results for sequential training, random selection and LITE-Adapt-∞ on this setting in Table 8. We see up to 18% forgetting rate in the sequential setting, which comes down to 5.6% and 2.3% with random pruning and Adapt-∞ pruning. The average accuracy is highest with our method at the final time step. These results suggest that lifelong learning from multi-task instruction tuning datasets is a significant problem in the language-only domain as well. Our proposed method can significantly alleviate forgetting of skills over time in the language domain, as well as preserve the maximum accuracy that can be achieved from each dataset.

---

[5]https://huggingface.co/datasets/YeungNLP/firefly-train-1.1M

