# OpenReview forum: "Adapt-$\infty$: Scalable Continual Multimodal Instruction Tuning via Dynamic Data Selection"
_ICLR.cc/2025/Conference — ICLR 2025 Poster_

### Official Review · Reviewer_eQ3c · 2024-10-30

**Soundness:** 2
**Presentation:** 3
**Contribution:** 3
**Rating:** 6
**Confidence:** 4

**Summary:**

This paper proposes a new scenario of lifelong instruction tuning, where Multimodal Large Language Models (MLLM) continuously learns from new datasets that include both new and redundant data samples. They then propose LAMP, a adaptive data selection approach designed for this context. The authors claims that the proposed method can select beneficial samples to adapt the current model's knowledge to the continuously changing dataset distribution.

**Strengths:**

- The setting of lifelong instruction tuning introduced in the paper is practical in real-world applications.
- The paper is overall well organized and easy to follow.
- The proposed method achieves convincing results in experiments, and comprehensive ablation studies are provided to validate the effectiveness of individual components.

**Weaknesses:**

- The method does not appear very novel to me, as both gradient-based clustering and ensemble scoring functions for data selection have been explored in previous works.
- In evaluation, the paper focuses on tasks with short answers but omits open-ended visual chat, where multimodal large language models (MLLMs) generate long responses. However, visual chat is essential for assessing the comprehensive capabilities of MLLMs.
- Compared to random pruning, which already serves as a strong baseline, the proposed method (including the LITE or efficient version) incurs significantly higher computational and time costs, and no efficiency analysis on the computational cost is provided.
- Clarification Issues. In Table 2, the authors do not clearly explain what "data size at *t*" refers to.

**Questions:**

- How is "accuracy" measured on LLaVA-Bench?
- In table 2, why does multi-task learning performs worse than random pruning which simply uses fewer data (e.g. accuracy 46.1% VS 47.2%) ? Can pruning more data lead to better result?
- Can you provide numerical comparisons on the computation cost of the proposed method against the random pruning baseline? Additionally, how does the computation cost of data selection compare with the training cost at step $t$?

---

> ### Author Response · Authors · 2024-11-22
>
> Dear Reviewer eQ3c,
>
> Thank you for reviewing our work, please see our response to the weaknesses and questions below:
>
> **Novelty**:
> We politely emphasize that our paper introduces three novel components in the research field.
>
> 1. Introducing the practical and realistic continual learning scenarios for MLLM instructional tuning: Lifelong Instruction Tuning (LiIT) (Lines 52-53 and Lines 72-75) with clarification of four critical challenges.
> 2. A new multi-way data selection approach for LiIT that can aggregate the benefits of various scoring functions based on the task.
> 3. A new scoring function i.e., image grounding score, that can distinguish visually-grounded data samples from the not-grounded samples.
>
> The main novelty of our work lies in our proposed lifelong instruction tuning scenario where *the model is trained on a multi-task instruction tuning at each time step*. This is in stark contrast to the conventional continual learning setup explored in most existing works and is a more practical variation of lifelong instruction tuning based on frequent data releases as seen in the research community today. This scenario brings on unique challenges that have not been tackled in previous continual learning works such as overlapping tasks, redundant data, rare tasks, etc. Our experiments range from frequently used baselines such as multi-task learning, random selection, and scoring-based selection to more sophisticated methods based on high-dimensional representations, yielding insightful results on the nature of lifelong learning in LLMs. Our proposed method is based on systematic insights from these experiments; further, the two-step process of task-based selection and semantic deduplication has not been explored in previous works. Most importantly, we show significant improvements with our method and establish a strong baseline for the scenario of lifelong instruction tuning.
>
> At the same time, we respectfully note that other reviewers (```9s1q, bvXZ, VSGx```) agree with the novelty and significance of our proposed scenario. For instance, Reviewer ```9s1q``` stated, *“...the paper reflects the authors' deep understanding of the field and direction, providing readers with extensive knowledge and unique insights, ..., opening up new avenues for future research.”* Additionally, these reviewers recognized that the proposed method appropriately and logically addresses well-structured challenges in lifelong multimodal instruction tuning scenarios, emphasizing the significance and uniqueness of the proposed approach compared to prior methods.
>
> The authors hope that the reviewer eQ3c understands our significance and practicality (often considered another ‘novelty’ from a scientific perspective) in formulating and addressing lifelong multimodal instruction tuning for the real world.
>
> **Evaluation on Visual Chat**: Thank you for the great suggestion, we have included examples of visual chat in our paper for our models (Figure 6, 7, and 8). Since there is no formal evaluation benchmark or quantitative metrics for multi-turn dialogue with MLLMs, we used a single representative example for qualitative analysis of visual chat results in the updated draft.
>
>  Figure 6 shows the multi-chat skills of the LLaVA model at $t$=0. In Figure 7, we note that the models trained using sequential learning (left) and random pruning (right) have poor retention of the multi-chat skill and reply with few-word and/or inaccurate answers. However, our method (results in Figure 8) retains this skill because it pools the multi-chat training data samples into a distinct cluster during the pseudo-task clustering step and selects sufficient samples from this cluster for training at each time step.
>
> **Efficiency Analysis**: Thank you for the great suggestion, we have added this to the paper. We present a comparison of the time taken by various data selection methods for our experimental setting (for training with 25k samples at each time step) in Table 6 in the Appendix (see revised pdf). Results are presented for 8 A100 GPUs. The total time taken to train the model without any methodical data selection (i.e., random pruning) is approximately 21 hours. Scoring-based selection methods generally require a forward pass of the model to extract the score (e.g., EL2N, entropy), which takes nearly 48 hours on 8A100 GPUs for all datasets in our experiments. Methods that use feature embeddings (e.g., COINCIDE) takes a similar amount of time since they uses a forward pass to extract hidden layer outputs from the models as data representations. Our proposed method requires longer time i.e., 92 hours, to perform a backward pass over the model and extract gradients, similar to LESS - which is a state-of-the-art method for data selection for targeted instruction tuning. However, with Lite-Lamp, we are able to significantly reduce this time due to systematic compression of the dataset at each timestep.

---

> > ### Author Response · Authors · 2024-11-22
> >
> > **Computational Complexity of Data Selection**: Random pruning is a rather effective method when it comes to data selection, as reported in many prominent works in the data selection literature. For instance, [1] reports that selecting 5% instruction tuning data via random pruning results in performance that is very close to that with 100% data; their proposed method i.e., gradient-based similarity results in a few points improvement over random. [2] show that it is hard to beat random pruning in very high pruning settings for the ImageNet dataset. When the availability of computational resources is low, random pruning is indeed a promising tradeoff for gain vs. compute. Nevertheless, data selection methods, including LAMP, are worth investigating for several reasons:
> >
> > - Selecting representative and important data samples can accelerate training [3, 4]. Besides, it is important from the perspective of scientific research as to why one data subset works better than another subset, selected randomly or otherwise. Developing data selection methods that are guided by concrete and intuitive hypotheses is the best way to understand this science.
> >
> > - Data selection methods either require importance scores or high-dimensional representations of data samples in order to represent the data landscape correctly and incur computational costs in this process. Similarly, LAMP uses high-dimensional representations for selecting data correctly and as we show in Table 2, exceeds the performance of random pruning by >5%, yielding important insights about data selection for lifelong learning. More importantly, LAMP promotes the forward transfer of skills during lifelong learning (109% relative gain) whereas random selection falls short (95.3% relative gain). This suggests that random selection is sub-optimal in practical settings where tasks can be unbalanced, redundant, or rare. Such scenarios need more sophisticated techniques for effective learning.
> >
> > We adopt several techniques that significantly reduce the computational complexity of our method such as
> >
> > (i) random projections for reducing the memory footprint of high-dimensional vectors,
> >
> > (ii) MiniBatch k-Means for clustering of large datasets, and
> >
> > (iii) systematic compression of datasets at each time step using deduplication (as outlined in Lite-Lamp, Section 4.4).
> >
> > LAMP can be made further optimized using libraries like faiss that quantize the high-dimensional space for fast nearest neighbor computations. Moreover, methods like LAMP will continue to reap benefits from faster inference algorithms. Thus, we think that LAMP is a computationally efficient approach to practical data selection for lifelong learning.
> >
> > **Multi-task vs. Random**: Multi-task pruning performs worse than random because the tasks are severely unbalanced across the datasets. For instance, there are nearly 600K samples for captioning and VQA, whereas there are only 20k samples for multilingual and referring expression comprehension. At larger data scales, the unbalanced task subsets lead to drops in performance for the rare tasks, however, in low data regimes (as demonstrated in random pruning), forgetting is less egregious even when the datasets are unbalanced. Especially for instruction tuning, using less but high-quality data does lead to better results, as also previously shown in [1, 5]. The scoring function-guided selection used in LAMP ensures that many of the high-quality data samples are being chosen consistently over time, and results in further improvements beyond random pruning.
> >
> > [1] Xia, Mengzhou, et al. "Less: Selecting influential data for targeted instruction tuning." arXiv preprint arXiv:2402.04333 (2024).
> >
> > [2] Zheng, Haizhong, et al. "Coverage-centric coreset selection for high pruning rates." arXiv preprint arXiv:2210.15809 (2022).
> >
> > [3] Goyal, Sachin, et al. "Scaling Laws for Data Filtering--Data Curation cannot be Compute Agnostic." Proceedings of the IEEE/CVF Conference on Computer Vision and Pattern Recognition. 2024.
> >
> > [4] Evans, Talfan, et al. "Data curation via joint example selection further accelerates multimodal learning." arXiv preprint arXiv:2406.17711 (2024).
> >
> > [5] Zhou, Chunting, et al. "Lima: Less is more for alignment." Advances in Neural Information Processing Systems 36 (2024).
> >
> > **Clarifications**:
> > - *Data size at $t$*: This refers to the size of the selected dataset for training at each time step $t$.
> > - *Accuracy of LLaVA-Bench*: This is computed using accuracy from an LLM as judge (GPT-4 in our experiments), as recommended in the original paper.

---

> > > ### Comment · Reviewer_eQ3c · 2024-11-22
> > >
> > > Thank you for the clarifications and the additional experiments, which address most of my concerns. Besides, the additional visual chat example effectively demonstrates the proposed method's applicability and effectiveness in real-world MLLM scenarios. I do agree that the proposed multimodal lifelong instruction tuning framework is practical and appreciate the improvements achieved by the authors. Therefore, I'd be willing to raise the score to lean towards accept.

---

> > > > ### Author Response · Authors · 2024-11-25
> > > >
> > > > Dear Reviewer eQ3c,
> > > >
> > > > Thank you for taking the time to review and discuss our responses and revisions in detail. We are grateful for these thoughtful feedback and discussions.
> > > >
> > > > We truly appreciate your support and the updated score!
> > > >
> > > > Best,
> > > >
> > > > Authors

---

### Official Review · Reviewer_VSGx · 2024-11-06

**Soundness:** 3
**Presentation:** 3
**Contribution:** 3
**Rating:** 6
**Confidence:** 5

**Summary:**

This paper introduces a novel approach, LAMP, for optimizing Lifelong Instruction Tuning (LiIT) of Multimodal Large Language Models (MLLMs). Traditional visual instruction datasets, which are frequently redundant and task-specific, hinder MLLMs' ability to continuously adapt and learn new skills effectively. LAMP addresses this by dynamically selecting and pruning data to maximize training efficiency while minimizing computational demands. It groups data into pseudo-skill clusters, applying adaptive selection to prioritize relevant samples for each skill and using a cluster-wise pruning method to reduce redundancies. This approach mitigates catastrophic forgetting, enhances knowledge transfer, and improves performance on rare tasks, all while leveraging only a fraction of the original dataset. LAMP's adaptive tuning demonstrates significant benefits for sustaining long-term model growth in evolving multimodal learning environments.

**Strengths:**

1. The proposed method mitigates catastrophic forgetting and supports forward transfer, enabling the model to retain skills and adapt to new tasks seamlessly.
2. LAMP’s cluster-wise pruning manages dataset growth by removing redundancies, keeping training scalable and resource-efficient.
3. This paper is well-writing and its organization is logical.

**Weaknesses:**

1. A primary concern regarding the proposed sequential data structure and the lifelong-based instruction tuning strategy is their practical applicability, as the current MLLMs typically rely on the integration of diverse tasks together for effective multitask training instead of sequential learning.
2. The paper mentioned that (lines 301-302) "our proposed multi-way approach can be seamlessly extended with new scoring functions based on users’ needs", however, the proposed method only employs four scoring functions in practice and lacks ablation experiments for additional scoring functions. I suggest the authors conduct related experiments to demonstrate the proposed method’s flexibility and scalability.
3. In Tab.2, the effectiveness of the random pruning strategy is notably highlighted. The computational costs and complexities associated with other sophisticated pruning techniques do not appear to correspond proportionately to the performance improvements achieved. This raises questions regarding the efficiency of the proposed pruning methods in comparison to simpler alternatives.
4. In Tab.2, although LITE-LAMP utilizes 4 times less training data (25k vs. 100k), its relative improvement dropped significantly when compared to LAMP, i.e., 99.7 vs. 109.7.

**Questions:**

Please focus on Weaknesses, and I encourage the authors to provide further discussion and clarification on these points.

---

> ### Author Response · Authors · 2024-11-21
>
> Dear Reviewer VSGx,
>
> Thank you for reviewing our paper and recognizing the merits of our work. Please see our response to the weaknesses below:
>
> **Sequential Learning**: We wholeheartedly agree that sequential learning is not a practical approach since most LLMs and MLLMs rely on multi-task instruction tuning from a massive dataset. Our work is motivated by this very fact, and hence, in contrast to most continual learning works that train the model on a single task at each time step, *we train our model on multi-task datasets at each timestep*. We consider the scenario where multiple multi-task instruction tuning datasets are available for training and there is significant overlap between tasks as well as some rare tasks that appear in one or more datasets only. This scenario is more practical because we see frequent releases of new multi-task instruction tuning datasets in the research community. Our experimental setup and proposed method enable the learning of new tasks and reinforcing as well as forward transfer of old tasks without having to retrain the model from scratch. We have emphasized and expanded these points in the revised draft (see the blue text in the Introduction).
>
>
>
> **Computational Complexity of Data Selection**: Random pruning is a rather effective method when it comes to data selection, as reported in many prominent works in the data selection literature. For instance, [4] reports that selecting 5% instruction tuning data via random pruning results in performance that is very close to that with 100% data; their proposed method i.e., gradient-based similarity results in a few points improvement over random. [5] show that it is hard to beat random pruning in very high pruning settings for the ImageNet dataset. When the availability of computational resources is low, random pruning is indeed a promising tradeoff for gain vs. compute. Nevertheless, data selection methods, including LAMP, are worth investigating for several reasons:
>
> - Selecting representative and important data samples can accelerate training [6, 7]. Besides, it is important from the perspective of scientific research as to why one data subset works better than another subset, selected randomly or otherwise. Developing data selection methods that are guided by concrete and intuitive hypotheses is the best way to understand this science.
> - Data selection methods either require importance scores or high-dimensional representations of data samples in order to represent the data landscape correctly and incur computational costs in this process. Similarly, LAMP uses high-dimensional representations for selecting data correctly and as we show in Table 2, exceeds the performance of random pruning by >5%, yielding important insights about data selection for lifelong learning. More importantly, LAMP promotes the forward transfer of skills during lifelong learning (109% relative gain) whereas random selection falls short (95.3% relative gain). This suggests that random selection is sub-optimal in practical settings where tasks can be unbalanced, redundant, or rare. Such scenarios need more sophisticated techniques for effective learning.
> - We adopt several techniques that significantly reduce the computational complexity of our method such as
>
>     (i) random projections for reducing memory footprint of high-dimensional vectors,
>
>     (ii) MiniBatch k-Means for clustering of large datasets, and
>
>     (iii) systematic compression of datasets at each time step using deduplication (as outlined in Lite-Lamp, Section 4.4).
>
> LAMP can be made further optimized using libraries like faiss that quantize the high-dimensional space for fast nearest neighbor computations.  Moreover, methods like LAMP will continue to reap benefits from faster inference algorithms. Thus, we think that LAMP is a computationally efficient approach to practical data selection for lifelong learning.
>
> **LAMP vs. Lite-LAMP**: The relative performance drops due to the reduction in budget size. Larger data budget enables more diversity in the data that is selected for training the model in the next step whereas a reduction in data size leads to lesser diversity and smaller improvements. It is notable that in spite of the significantly lower data budget in Lite-LAMP, our method mitigates catastrophic forgetting at all timesteps and preserves ~100% relative gain.

---

> > ### Author Response · Authors · 2024-11-21
> >
> > **Additional Scoring Functions**: Thank you for the suggestion regarding the ablations of scoring functions used in our experiments. The scoring functions we have chosen, i.e., Image Grounding score, EL2N, Entropy, and Perplexity, are based on promising preliminary results on the individual score-selection methods in our experiments (see corresponding rows in Table 2) as well as existing literature [1, 2]. These score functions have been shown to perform well for tasks such as image classification (EL2N, entropy) and language modeling (perplexity). To conduct ablations on our multi-way setup, we ran the following experiments:
> >
> > (a) *Additional scoring functions*: We added GraND [2] and AUM [3] to the set of score functions (i.e., a total of 6 functions) for LAMP.
> >
> > (b) *LAMP without Image-Grounding score*: We are running an experiment using only EL2N, Entropy, and Perplexity as the set of scoring functions.
> >
> > From (a), we did not see any improvements over our best results with LAMP using four scoring functions because GraND and AUM functions did not score high in entropy for any of the pseudo-task clusters. We will report on results from (b) as soon as they are available. From these partial results, we can conclude that a score function is useful only when it is highly discriminative for a task cluster. We will add these additional results and discussion to the camera-ready version of the version.
> >
> >
> > [1] Marion, Max, et al. "When less is more: Investigating data pruning for pretraining llms at scale." arXiv preprint arXiv:2309.04564 (2023).
> >
> > [2] Paul, Mansheej, Surya Ganguli, and Gintare Karolina Dziugaite. "Deep learning on a data diet: Finding important examples early in training." Advances in neural information processing systems 34 (2021): 20596-20607.
> >
> > [3] Pleiss, Geoff, et al. "Identifying mislabeled data using the area under the margin ranking." Advances in Neural Information Processing Systems 33 (2020): 17044-17056.
> >
> > [4] Xia, Mengzhou, et al. "Less: Selecting influential data for targeted instruction tuning." arXiv preprint arXiv:2402.04333 (2024).
> >
> > [5] Zheng, Haizhong, et al. "Coverage-centric coreset selection for high pruning rates." arXiv preprint arXiv:2210.15809 (2022).
> >
> > [6] Goyal, Sachin, et al. "Scaling Laws for Data Filtering--Data Curation cannot be Compute Agnostic." Proceedings of the IEEE/CVF Conference on Computer Vision and Pattern Recognition. 2024.
> >
> > [7] Evans, Talfan, et al. "Data curation via joint example selection further accelerates multimodal learning." arXiv preprint arXiv:2406.17711 (2024).

---

> > > ### Author Response · Authors · 2024-11-26
> > > **Additional Results and Gentle Reminder**
> > >
> > > Dear Reviewer VSGx,
> > >
> > > We have results for experiment (b) for **Additional scoring functions** mentioned in our previous response. The results indicate that the multi-way pruning module in LAMP is effective without our Image-Grounding score too.
> > >
> > > |Methods|Relative Gain|Forgetting rate|Avg. Accuracy|
> > > |---|---|---|---|
> > > | IG Score only | 92.3 | 5.6 | 45.6 |
> > > |(**new**) LAMP (without IG Score) | 106.9 | 0.3 | 51.2 |
> > > |LAMP (all 4 scoring functions)| 109.7 | 0.4 | 52.5|
> > >
> > > Evidence shows that the multi-way pruning method is very effective when multiple *good* scoring functions are candidates. This also suggests that the performance with our method can improve with better scoring functions, making LAMP a versatile method. We will add these ablations to the camera-ready version of the paper.
> > >
> > > In summary, we have discussed the practicality of our lifelong instruction tuning setting and demonstrated the efficiency and generalizability of our proposed LAMP method (please see additional results in the updated pdf). Thank you for your time and effort in reviewing our paper and for your constructive feedback, which has significantly contributed to improving our work. We hope the added clarifications and the revised submission address your concerns and kindly request to further reconsider the rating/scoring. We are happy to provide further details or results if needed.
> > >
> > > Warm Regards,
> > >
> > > Authors

---

> ### Author Response · Authors · 2024-12-01
> **We have less than two days left in the discussion period.**
>
> Dear Reviewer VSGx,
>
> We sincerely appreciate your efforts in reviewing our paper and your constructive comments. Since there are less than two days left in the discussion period, could you please read our responses to check if your concerns are clearly addressed? We believe that our responses resolved all your concerns.
>
> We understand that the criteria for rating a paper can sometimes be subjective; however, if you agree that our work does not have remaining major concerns, we would like to kindly suggest your re-evaluation of the initial rating of this submission.
>
> Please let us know if you have any remaining questions, and we will be more than happy to address them.
>
> Best,
> Authors

---

### Official Review · Reviewer_bvXZ · 2024-11-06

**Soundness:** 3
**Presentation:** 4
**Contribution:** 3
**Rating:** 8
**Confidence:** 4

**Summary:**

This paper introduces Lifelong Multimodal Instruction Tuning, which differs from continual Multimodal Instruction Tuning in that the previous datasets are still included in the dataset pool. To effectively select the data at each timestep, the authors propose a framework that adaptively  selects data samples based on their importance and data balance. The experiments show that the method is more effective than baselines.

**Strengths:**

- The considered lifelong setting is more practical than the traditional continual learning setting for multimodal LLMs, since currently the most important aspect of large models is to use all data available better.
- The proposed framework considers the key difficulties of data sampling in adapting multimodal LLMs with progressively growing datasets and reasonably tackles the problem with carefully designed pipelines. I find sections 3 and 4 meaningful with rigorous thoughts and experiment analysis.
- The experiments are thorough and clearly show the effectiveness of the proposed framework over the considered baselines.
- The paper is written very well.

**Weaknesses:**

- The proposed method, except for the image grounding score, seems not to be specifically designed for multimodal LLMs and can be potentially applied to broader domains like pure LLMs. However, the paper only considers multimodal LLMs which lowers the impact and importance of the paper.

**Questions:**

the t-SNE visualization has been shown to have high stochasticity across seeds. Have you manually tuned for your method and other baselines throughout the paper?

---

> ### Author Response · Authors · 2024-11-21
>
> Dear Reviewer bvXZ,
>
> Thank you for reviewing our work and appreciating the rigorous thoughts and experimental analysis that went into the paper. Please see our response to the weaknesses below:
>
> **LAMP for LLMs**: Thank you for the suggestion, we agree that the data selection method designed in our paper can be extended to LLMs as well, and will be exciting future work. During the rebuttal period, we have decided to add experiments using LAMP on LLMs only. One thing to note is that LLMs generally undergo multiple post-training phases after instruction tuning, such as alignment and other fine-tuning stages. On the other hand, our method and the proposed lifelong instruction tuning scenario focus on the instruction tuning phase, and it might be important to understand how data selection interacts with these subsequent post-training phases, which will be highly practical and exciting future research direction in lifelong learning of LLMs with data selection.
>
> Our experimental setup is as follows:
>
> *Model*: LLaMA3-8B
>
> *Datasets in the order of training*:
> - Natural Instructions [~1600 NLP tasks]
> - Alpaca CoT [geared towards Chain-of-Thought ability]
> - xP3 [Multilingual]
> - UltraChat [Multi-turn dialogues]
> - Firefly [Chinese only]
>
> Since training and evaluation takes substantial time, we plan to report our results on this experimental setup by this Friday.
>
> **t-SNE Seed**: We have used the t-SNE visualizations to only understand the spatial distribution of feature vectors from different sources. We repeated the t-SNE visualization experiment across 10 seeds and did not find a significant difference in the quality of clusters across these seeds for any of the feature vector sources; the results reported in our paper are stable across seeds. The figures reported in the paper are generated from the same seed for different datasets for fair comparison.

---

> ### Author Response · Authors · 2024-11-24
>
> We have results from our runs with LAMP and LLaMA3-8B with the datasets enumerated in our earlier response i.e., Natural Instructions, Alpaca CoT, xP3, UltraChat, and Firefly. We use the set of evaluation benchmarks used in [1] to evaluate our models (MMLU, GSM, BBH, TydiQA, CodexEval, AlpacaEval) and compute the following metrics based on performance across these benchmarks: Average accuracy, Relative gain, and Forgetting rate (see  Sec. 5.1 in our paper). The results at each timestep are as follows (numbers indicate average accuracy):
>
>
> |Methods|NI (t=1)|Alpaca (t=2)|xP3 (t=3)|UltraChat(t=4)|Firefly(t=5)|
> |---|---|---|---|---|---|
> |Sequential|35.1|**43.9**|32.1|41.8|36.8|
> |Random|35.1|42.7|40.1|42.9|41.8|
> |Ours|35.1|$\underline{43.3}$|**42.6**|**43.6**|**43.1**|
>
>
> Based on these results, the relative gain and forgetting rate are as follows:
>
> |Methods|Relative Gain|Forgetting rate|Avg. Accuracy|
> |---|---|---|---|
> |Sequential| - | 18.1 |36.8 |
> |Random| 91.6 | 5.6 |41.8 |
> |Ours| **97.8** | **2.3** | **43.1**|
>
> The model is trained on 25k samples at each time step, similar to the MLLM experiments in our paper. We implemented the Lite-Lamp version of our method for these experiments. We see up to 18% forgetting rate in the sequential setting, which comes down to 5.6% and 2.3% with random pruning and LAMP pruning. The average accuracy is highest with our method at the final time step. These results suggest that:
>
> (a) lifelong learning from multi-task instruction tuning datasets is a significant problem in the language-only domain as well and
>
> (b) our proposed method can significantly alleviate forgetting of skills over time, as well as preserve the maximum accuracy that can be achieved from each dataset.
>
> [1] How Far Can Camels Go? Exploring the State of Instruction Tuning on Open Resources

---

> ### Author Response · Authors · 2024-11-25
> **A gentle reminder**
>
> Dear Reviewer bvXZ,
>
> Thank you for your effort in reviewing our paper. We kindly notify you that the end of the discussion stage is approaching. Could you please read our responses to check if your concerns are clearly addressed? During the rebuttal period, we made every effort to address your concerns faithfully:
>
> - We have additionally demonstrated strong generalizability of the proposed LAMP in the language domain.
> - We have clarified the reviewer's question regarding the relevance between t-SNE stochasticity and model performance.
>
> Thank you for your time and effort in reviewing our paper and for your constructive feedback, which has significantly contributed to improving our work. We hope the added clarifications and the revised submission address your concerns and kindly request to further **reconsider the rating/scoring**. We are happy to provide further details or results if needed.
>
> Warm Regards,
> Authors

---

> > ### Comment · Reviewer_bvXZ · 2024-11-27
> >
> > Thanks for the detailed reply. The authors have adequately addressed my concerns. I thus increase the score. I suggest authors add the additional LLM experiments to the appendix upon publication.

---

> > > ### Author Response · Authors · 2024-11-27
> > >
> > > Thanks for your continued engagement and for increasing your score. We are glad we adequately addressed your concerns, and we will definitely add the additional LLM experiments to the appendix upon publication; we believe this result further strengthens the generalizability and effectiveness of our proposed method.

---

### Official Review · Reviewer_9s1q · 2024-11-09

**Soundness:** 3
**Presentation:** 3
**Contribution:** 3
**Rating:** 6
**Confidence:** 4

**Summary:**

This is an interesting article that addresses the issue of forgetting in multimodal large models within multi-task and continual learning scenarios from the perspective of instruction dataset selection and proposes effective strategies. Traditional training methods often lead to the model forgetting previously learned skills when new tasks are introduced. However, the article introduces the LAMP (Lifelong and Adaptive Multi-way Pruning) method, which uses gradient vectors for pseudo-task clustering and combines an entropy-maximization strategy for sample selection, achieving effective sample filtering and diversity balance. LAMP employs semantic redundancy detection and coverage-based sampling strategies to prune the data pool, control its size, and prevent unrestrained data pool expansion, while ensuring the representativeness and diversity of tasks. Moreover, training budgets are allocated reasonably across task clusters, allowing the model to be sufficiently trained on each task, thereby reducing forgetting and enhancing generalization capabilities. Overall, LAMP performs well in multi-task and continual learning environments, ensuring that the model retains previously learned skills when training on new tasks, even with limited computational resources, thus achieving efficient and balanced continual learning.

**Strengths:**

This article starts from the current situation where the instruction-tuning datasets used in fine-tuning multimodal large models are continuously increasing and points out the redundancy problem among these datasets. It then proposes a solution to address the issue of continual learning, demonstrating significant practical relevance. In the design of the proposed solution, the authors first analyze the problems and limitations of existing methods, then detail how the new solution effectively addresses these issues, showcasing clear research objectives and a logical structure. The narrative of the paper reflects the author’s deep understanding of the field and direction, providing readers with extensive knowledge and unique insights. Moreover, the article verifies the correctness and validity of the proposed solution through extensive experiments, opening up new avenues for future research and demonstrating considerable i

**Weaknesses:**

Although it is good to see the paper as a whole, there still exist some minor drawbacks in this work.
1.	In Figure 2, the meanings of the axes in subfigure C are unclear, and the explanatory content for subfigure A is insufficient, failing to adequately explain the issue that the figure is intended to illustrate.
2.	The text descriptions in all the images are not very clear and appear somewhat distorted.
3.	In the paper, clustering operations on datasets based on gradients and subsequent pruning of redundant data use k-means and pair-wise cosine similarity, respectively. This computational approach likely incurs significant time and space costs in such large-scale instruction datasets. The question arises as to whether there are corresponding measures to address these challenges.
4.	During the data selection process, a pool of function tools is used, and the function that produces higher average entropy is ultimately chosen. In this process, combining multiple functions can be considered to achieve more robust data selection.

**Questions:**

please refer to the weakness part

---

> ### Author Response · Authors · 2024-11-21
>
> Dear Reviewer 9s1q,
>
> Thank you for recognizing the merits of our work. Please see our response to the weaknesses below:
>
> **Figures**: Thank you for the suggestions for making the figures more understandable. We have updated the explanations for subfigures A and C in Figure 2 and improved the text font in all of our figures for clarity in the updated version of our paper (see revision).
>
> **Computational costs of clustering in LAMP**: We note that our clustering approach incurs only marginal additional computational cost and does not require significant resources. Naive k-means clustering has a computational complexity of $O(tknd)$ where $t$ = iterations of the algorithm, $k$ = number of clusters, $n$=number of samples, $d$=dimension of vectors, implying that the computation costs rise linearly with each of these factors.
>
> - The value $t$ is constant in our experiments.
> - The variables $k$ and $d$ are small in our experiments (see Lines 354-358). The optimal values of $k$ are between 5 and 50, and the value of $d$ is capped at 8192 by virtue of using random projections to compress high-dimensional vectors (Lines 877-881 in Appendix).
> The value of $k$ can increase with the size of the datasets, however, there exist many efficient methods for calculating pairwise cosine similarities effectively. We use the efficient implementation of `MiniBatch-KMeans` in scikit-learn to compute clusters for our datasets, which significantly reduces the compute time.
> - The maximum time taken for a single k-means clustering run in our LAMP experiments using this implementation is 48 minutes on a 48-core CPU for approximately 1.8 million samples. In addition, the `faiss` library can also be used for efficient computation of pairwise cosine similarities using quantization.
> - Further, our proposed approach `Lite-Lamp` seeks to reduce the number of samples $n$ at each step, effectively reducing the time taken for k-means clustering. The maximum time taken for a single k-means clustering run in Lite-LAMP experiments is **15 minutes** on the 48-core CPU for approximately 660K samples, which makes it **as efficient as other data selection methods** like LESS [1] and COINCIDE [2].
>
> Since there are many ways to control each of the variables that contribute to the complexity of k-means clustering ($k$, $n$ and $d$), we think that LAMP is not a prohibitively computationally expensive method.
>
> [1] Xia, Mengzhou, et al. "Less: Selecting influential data for targeted instruction tuning." arXiv preprint arXiv:2402.04333 (2024).
>
> [2] Lee, Jaewoo, Boyang Li, and Sung Ju Hwang. "Concept-skill Transferability-based Data Selection for Large Vision-Language Models." arXiv preprint arXiv:2406.10995 (2024).
>
> **Combinatorial Prediction of Function Tools for Sample Selection**: Each function tool assigns a different value to every sample in our dataset. Since each functional tool has its unique range and distribution, we normalize these values to enable comparison of entropies across these tools. We appreciate the suggestion for combining these distributions to enable further gains for our method. We believe this idea presents an exciting direction for future work, potentially improving the robustness and efficiency of data selection in future implementations, but also introduces additional challenges that need to be addressed:
>
> 1) determining top-$k$ or dynamic number of beneficial function tools for each data pair, and
> 2) controlling the relative influence of predictions.
>
> The second point would be crucial as different tools may have varying degrees of relevancy and reliability for different samples, and an imbalance could affect the robustness of the selection process.
>
> In summary, the authors appreciate the reviewer for raising this constructive discussion and believe that exploring these challenges, as well as developing methods to effectively combine the outputs of multiple functional tools, will be a meaningful direction for future work.

---

> ### Author Response · Authors · 2024-11-25
> **A gentle reminder**
>
> Dear Reviewer 9s1q,
>
> Thank you for your effort in reviewing our paper. We kindly notify you that the end of the discussion stage is approaching. Could you please read our responses to check if your concerns are clearly addressed? During the rebuttal period, we made every effort to address your concerns faithfully:
>
> - We have updated the explanations for subfigures A and C in Figure 2 and improved the text font in all of our figures
> - We provide the details regarding the computational costs of clustering in LAMP. We would like to note that the proposed approach is as efficient as other data selection methods like LESS [1] and COINCIDE [2].
> - We further provide a discussion regarding the Combinatorial Prediction of Function Tools for Sample Selection, which should be a promising research direction that entails two distinct and meaningful research challenges.
>
> Thank you for your time and effort in reviewing our paper and for your constructive feedback, which has significantly contributed to improving our work. We hope the added clarifications and the revised submission address your concerns and kindly request to further **reconsider the rating/scoring**. We are happy to provide further details or results if needed.
>
> Warm Regards,
> Authors

---

> ### Author Response · Authors · 2024-12-01
> **We have less than two days left in the discussion period.**
>
> Dear Reviewer 9s1q,
>
> We sincerely appreciate your efforts in reviewing our paper and your constructive comments. Since there are less than two days left in the discussion period, could you please read our responses to check if your concerns are clearly addressed? We believe that our responses resolved all your concerns.
>
> We understand that the criteria for rating a paper can sometimes be subjective; however, if you agree that our work does not have remaining major concerns, we would like to kindly suggest your re-evaluation of the initial rating of this submission.
>
> Please let us know if you have any remaining questions, and we will be more than happy to address them.
>
> Best,
> Authors

---

### Public Comment · ~Lai_Wei7 · 2024-11-19
**Thanks for your interesting work**

Hope that InstructionGPT-4 (https://arxiv.org/abs/2308.12067), which also focuses on multimodal data selection, can be discussed or included in a part of your paper :)

---

### Author Response · Authors · 2024-11-21
**General Comments**

We thank the reviewers for their time and valuable comments. We appreciate that the reviewers have recognized:

- **Authors’ deep understanding of the field and direction**, providing readers with extensive knowledge and unique insights. [9s1q]

- The proposed Lifelong instruction learning setting is **more practical in real-world applications**. [bvXZ, eQ3c]

- **Clear research objectives and a logical structure**. [9s1q, VSGx]

- **The proposed method appropriately addresses the challenges** in the scenario. [bvXZ, VSGx]

- **Extensive experiments and valuable analyses**. [9s1q, eQ3c, bvXZ]

- **The paper is written very well; well organized, and easy to follow**. [bvXZ, eQ3c, VSGx]


During the rebuttal period, we made every effort to address all the reviewers' concerns faithfully.  We hope that our responses have addressed your comments. We thank you again for reviewing our work.

---

### Meta-Review · Area_Chair_Fz7F · 2024-12-19

**Metareview:**

This work investigates Lifelong Instruction Tuning (LiIT) via dynamic data selection, which can learn new skills while alleviating the forgetting rate. Before rebuttal, the reviews were mixed and the concerns were about novelty, generalization for LLMs, efficiency, etc. The discussion addressed most of the concerns. After rebuttal, all reviewers had the positive rating due to the practical settings and promising performance. Please incorporate suggested experiments and comments from reviewers in the revised submission.

**Additional Comments On Reviewer Discussion:**

Before rebuttal, the overall rating was mixed. After discussion, Reviewer bvXZ increased the score to 8 and Reviewer eQ3c changed the score to positive. All reviewers agree to accept the work.

---

### Decision · Program_Chairs · 2025-01-22

Accept (Poster)